Effects of groundwater abstraction on two keystone tree species in an arid savanna national park

Shadwell Eleanor
February Edmund edmund.february@uct.ac.za
Department of Biological Sciences, University of Cape Town , Private Bag, Rondebosch , South Africa
Ritsema Coen
Electronic publication date: 2017 Jan 25
Publication date: 2017
Volume: 5
Electronic Location ID: e2923
Received 2016 Sep 15; Accepted 2016 Dec 19
Copyright: ©2017 Shadwell and February
Copyright year: 2017
Copyright holder: Shadwell and February
License: This is an open access article distributed under the terms of the Creative Commons Attribution License, which permits unrestricted use, distribution, reproduction and adaptation in any medium and for any purpose provided that it is properly attributed. For attribution, the original author(s), title, publication source (PeerJ) and either DOI or URL of the article must be cited.
License URL: https://creativecommons.org/licenses/by/4.0/

Keywords: Acacia erioloba, Acacia haematoxylon, Water use, Water stress, Physiological stress, Canopy dieback, Tourism, Threshold for potential concern

Funding: Andrew Mellon Foundation of New York #30600716 The research was funded by the Andrew Mellon Foundation of New York #30600716. The funders had no role in study design, data collection and analysis, decision to publish, or preparation of the manuscript.

==============================
Background

In arid systems with no surface water, deep boreholes in ephemeral river beds provide for humans and animals. With continually increasing infrastructure development for tourism in arid wildlife parks such as the Kgalagadi Transfrontier Park in southern Africa, we ask what effects increased abstraction may have on large trees. Large trees in arid savannas perform essential ecosystem services by providing food, shade, nesting sites and increased nutrients for many other plant and animal species and for this are regarded as keystone species.

Methods

We determine seasonal fluctuations in the water table while also determining the water source for the dominant large tree species in the Auob and Nossob rivers in the Park. We also determine the extent to which these trees are physiologically stressed using leaf δ13C, xylem pressure potentials, specific leaf area and an estimate of canopy death. We do this both upstream and downstream of a low water use borehole in the Auob River and a high water use borehole in the Nossob River.

Results

Our results show that the trees are indeed using deep groundwater in the wet season and that this is the same water used by people. In the dry season, trees in the Auob downstream of the active borehole become detached from the aquifer and use more isotopically enriched soil water. In the Nossob in the dry season, all trees use isotopically enriched soil water, and downstream of the active borehole use stomatal regulation to maintain leaf water potentials. These results suggest that trees in the more heavily utilised Nossob are under more water stress than those trees in the Auob but that trees in both rivers demonstrate physiological adaptation to the changes in available water with smaller heavier leaves, no significant canopy dieback and in the dry season in the Nossob stomatal regulation of leaf water potentials.

Discussion

An increase in abstraction of groundwater particularly at the Nossob borehole may cause an additional draw down of the water table adding to the physiological stress demonstrated in our study. The managers of the Kgalagadi Transfrontier Park have a mandate that includes biodiversity conservation. To fulfil this mandate, upper and lower thresholds for groundwater abstraction that allow for an adequate ecological reserve have to be determined.

Introduction

All of the perennial rivers in the Kruger National Park in South Africa have been anthropogenically modified by water abstraction and increased sediment and pollution loads (Du Preez & Steyn, 1992; Rogers & Biggs, 1999). It has also been demonstrated that ephemeral rivers such as the Auob and the Nossob in the more arid Kgalagadi Transfrontier Park are currently not threatened by water abstraction even though the first boreholes were sunk in these rivers in the 1930’s (Mills & Retief, 1984; Nel et al., 2007; Van Wyk & Le Riche, 1984). Tourism development is seen as fundamental to community development and poverty alleviation in many parts of the world and is therefore heavily promoted in many National Parks (Binns & Nel, 2002; Neto, 2003). To this end, there have been substantial increases in tourist infrastructure at Nossob, Kieliekrankie and Urikaruus tourist camps in the Kgalagadi Transfrontier Park (SANParks, 2016) (Fig. 1). As there is no natural surface water in this park, all the water for this increase in tourism development has to come from groundwater abstraction (Mills & Retief, 1984; Van Wyk & Le Riche, 1984). The economic benefits of tourism have been illustrated in several publications, while the environmental impacts resulting from groundwater abstraction for this have also been demonstrated (Hall, 2001; Mbaiwa, 2003; Stromberg, 1993). Studies have shown that an increase in infrastructure development in arid environments can result in both short and long term water table declines with negative repercussions on groundwater dependant ecosystems (Barron et al., 2014; Groom, Froend & Mattiske, 2000; Lite & Stromberg, 2005). The effects of such abstraction on vegetation structure in arid systems with no surface water have however not been demonstrated.

Figure 1 Map of the Kgalagadi Transfrontier Park showing the location of the four boreholes at which both water and twig samples were collected.

The borehole for the Nossob Rest Camp is between the North Kwang and Qubit’je Quap boreholes with a pipeline down to Nossob.

Arid savanna such as at our study site in the Kgalagadi Transfrontier Park occur across the world, from the Caatinga in northeast Brazil, through to the Mulga of central Australia (Huntley & Walker, 2012; Nix & Austin, 1973). In arid savanna trees are scattered in an expansive bare ground, shrub, and grass matrix. At our study site Acacia erioloba and Acacia haematoxylon dominate in the dry river beds as the only large tree species, along with scattered smaller trees and bushes with dwarf shrubs and grasses occurring in sparse clumps (Leistner, 1959; Van Rooyen et al., 2008). Both A. erioloba and A. haematoxylon provide nesting sites, shade, food resources and soil nutrients for a variety of other plants and animals and because of this are considered as keystone species (Dean, Milton & Jeltsch, 1999; Milton & Dean, 1995).

Recent research has demonstrated that trees growing in the Kuruman River a system similar to the Auob and the Nossob are using deep (56 m) water (Schachtschneider & February, 2013). If these trees are indeed reliant on deep water and this is the same water that is being pumped for tourism development, then there is an urgent need to develop a conceptual understanding of groundwater recharge processes and the interaction between groundwater and the trees growing in the river bed. Several studies have now demonstrated that it is possible to determine the water source of a plant through an analysis of the stable hydrogen and oxygen isotope ratios of the water in the xylem tissue of that plant (Dawson & Ehleringer, 1991; Ehleringer & Dawson, 1992; February et al., 2007). The method is based on the assumption that the water in the xylem of non-photosynthesising tissue will have the same stable isotope ratio as the water source (Dawson & Ehleringer, 1991). We determine the interaction between groundwater and the two tree species A. erioloba and A. haematoxylon growing in the dry beds of the ephemeral Auob and Nossob rivers in the Kgalagadi Transfrontier Park (Fig. 1). We do this by firstly determining the water source for the two species using hydrogen and oxygen isotope ratios of both xylem and borehole water (Schachtschneider & February, 2013). We then determine the extent to which the trees are physiologically stressed using stable carbon isotope ratios of the leaves, midday xylem pressure potentials, specific leaf area and an estimate of canopy death (Farquhar et al., 1989; Liu & Stützel, 2004; Scholander et al., 1965).

National Parks in South Africa have adopted strategic adaptive management with clear ecosystem management goals. The achievement of these goals is evaluated by using environmental indicators (Biggs & Rogers, 2003). With a constant demand for resources in the Kgalagadi Transfrontier Park there is a real need to ascertain what the natural balance of groundwater is and what the interaction is between this water and the trees growing in the dry river beds. Such an understanding is critical for determining the influence of water table flux on riparian species thereby allowing park management to develop a sustainable water resource allocation that meets the needs of both tourism and the environment (Chen et al., 2016).

Methods

Study site

The study site is located in the western and southern part of the Kgalagadi Transfrontier Park, in southern Africa. We sampled at four boreholes in the Park, Urikaruus waterhole and Kamqua waterhole upstream and downstream respectively of the Urikaruus Wilderness camp borehole (−26.010832°, 20.349627°) in the Auob River; and North Kwang and Qubit’je Quap boreholes upstream and downstream respectively of the Nossob camp borehole (−25.287383°, 20.537513°) in the Nossob River (Fig. 1).

The climate for the region is characterised by distinctly seasonal rainfall with a hot wet season from late November to early April. Mean annual rainfall (1984–2014) increases from 180 mm at Nossob (−25.4212°, 20.5968°) in the north to 220 mm at Twee Rivieren 116 km to the south (−26.4721°, 20.6116°). Mean annual maximum and minimum temperatures at Twee Rivieren are 36.7 °C and 0.1 °C for January and July respectively (South African Weather Bureau).

The general landscape is primarily covered in aeolian dune sand underlain by silcretes and calcretes of the Cenozoic Kalahari Group. Our sampling sites in the riverbed, however, typically consist of fine grained silts (Mucina & Rutherford, 2006). In the Auob, A. erioloba and A. haematoxylon (biogeographically endemic to the Kalahari) are the dominant tree species, with Acacia mellifera and Rhigozum trichotomum common shrubs. Grasses such as the annual Schmidtia kalahariensis and perennial Stipagrostis obtusa occur in sparse clumps (Leistner, 1959; Mucina & Rutherford, 2006). The vegetation in the Nossob is very similar to that of the Auob, but without A. haematoxylon, only large A. erioloba trees. Common grasses here are the perennial Panicum coloratum var. coloratum and Eragrostis bicolor (Bothma & De Graaff, 1973; Mucina & Rutherford, 2006).

Water source

Borehole water level

To monitor fluxes in water table through time piezometers (Solinst-Leverlogger, Georgetown, Ontorio, Canada) were inserted into our sampled boreholes, North Kwang and Qubit’je Quap in the Nossob and Urikaruus and Kamqua in the Auob from the 25th August 2012 to the 31st January 2014 (Fig. 1).

Oxygen and hydrogen stable isotopes

We use the hydrogen and oxygen stable isotope ratios of water extracted from non-photosynthesising xylem tissue to determine the water source for both A. erioloba and A. haematoxylon (February et al., 2007; Schachtschneider & February, 2013). The method is based on the assumption that water extracted from non-photosynthesising xylem tissue of a tree will have the same isotopic ratio as the source water for that tree (White, Cook & Lawrence, 1985). For this we collected two twig samples (∼0.5 cm × 6 cm) from six A. erioloba and six A. haematoxylon (twelve trees) upstream and six trees for each species (twelve trees) downstream of the active borehole for the Urikaruus Wilderness Camp (Fig. 1). At the same time we also collected two twig samples from eight A. erioloba trees upstream and eight trees downstream of the active borehole for the Nossob camp (Fig. 1). These samples were collected into borosilicate tubes (Kimax-Kimble, New Jersey, USA) which were placed directly onto a cryogenic vacuum extraction line to separate out the water for stable isotope analysis (Schachtschneider & February, 2013). We did this on three occasions, in January 2013 (wet season), July 2013 (dry season) and January 2014 (wet season).

We also collected water from each of our four boreholes at Urikaruus, Kamqua, North Kwang and Qubit’je Quap in August 2012 (dry season), November 2012 and January 2013 (wet season) July and August 2013 (dry season) and January 2014 (wet season). Rain water samples were collected opportunistically at Twee Rivieren, North Kwang and Nossob. All water samples were stored in 20 ml glass screw top bottles (Wheaton Liqid Scintillation Vials, Millville, NJ, USA) before analyses for both 2H/H and 18O/16O using a Thermo Delta Plus XP Mass Spectrometer (Hamburg Germany) at the University of Cape Town. The same mass spectrometer was used to determine 13C/12C ratios of the leaves from our study trees.

Plant moisture stress

Leaf stable carbon isotope ratios

We use the stable carbon isotope ratios of leaves to determine the efficiency of carbon assimilation and plant water use (intrinsic water use efficiency) for our study trees (Dawson et al., 2002; Farquhar et al., 1989). Plants regulate the amount of water lost to the atmosphere by closing or opening stomata. As stomata close with a decrease in available water there is less carbon assimilated and less discrimination against the heavier 13C isotope resulting in more positive leaf δ13C values (Farquhar et al., 1989). In the middle of the wet season in January 2013, we collected ten fully mature whole leaves from each of our study trees in both the Auob (twenty four trees) and the Nossob (sixteen trees) rivers. Prior to mass spectrometry, the leaves were oven dried at 70 °C to constant weight before being ground to a fine powder using a Retsch MM200 ball mill (Retsch, Haan, Germany).

Xylem pressure potentials

Xylem pressure potentials (XPP’s) are a relative indicator of the amount of water available to the plant through a determination of the amount of tension the water column is under (Hempson, February & Verboom, 2007; Scholander et al., 1965). We determined midday XPP’s in both the wet and dry season of 2013 using a Scholander-type pressure chamber (PMS Instrument Company, Corvallis, OR, USA). We specifically used midday rather than predawn XPP’s because of the hazards related to working in the dark with large carnivores present in the area (Hempson, February & Verboom, 2007). We also assumed that midday readings in the middle of both the wet and dry seasons should sufficiently illustrate any differences in plant available water between trees at our study site.

Specific leaf area

Specific leaf area (the ratio of leaf area to leaf dry mass) declines with decreasing soil moisture as leaves become smaller and heavier to reduce water loss and susceptibility to desiccation (Ackerly et al., 2002; Liu & Stützel, 2004). We collected ten fully mature whole leaves from each of our trees in both the wet and the dry season of 2013. The entire leaf including both petiole and rachis were then photographed against a white background before determining leaf area using the open source software ImageJ (Abràmoff, Magalhães & Ram, 2004).

Canopy dieback

Using two photographs for each tree we determined the amount of canopy dieback on all of our study trees. Semi-deciduous species such as A. erioloba and A. haematoxylon are rarely leafless in the dry season due to synchronised leaf fall and new leaf emergence (Sekhwela & Yates, 2007). Twigs and branches with no leaves were therefore attributed to heavy browse or drought induced leaf mortality. We photographed the canopy of each of our study trees in both the wet and the dry season of 2013. Each photograph was taken through an 18–200 mm f/3.5–6.3 HSM DC lens (Sigma, Fukushima, Japan) with PRO1 D UV (W) filter attached, (Kenko, Tokyo, Japan) fixed at 52 mm, F8 aperture using a Nikon D60 camera (Nikon, Ayuthaya, Thailand) set on a tripod (290 Series; Manfrotto, Cassola, Italy) 1.5 m off the ground. The distance between camera and tree was measured from the mid-point of the tripod to the base of the tree. Photographs were taken between 10h00 and 16h00 in two directions, east to west and north to south, unless obstructed when the direction was switched through 180° . The angle of the tripod head was adjusted using the spirit level set into the tripod, so that the camera was always horizontal (relative to the ground) and tilted vertically keeping the entire tree canopy just inside the field of view. For the calibration of each photograph, a retractable 5 m aluminium ranging rod (levelling staff; Leica Geosystems, St. Gallen, Switzerland) was held vertically at the edge of the canopy in the field of view (Fig. S1).

Each photograph was overlaid by a grid (50 cm boxes, subdivided by 10), using Adobe Photoshop (CS5 v12.0 × 32© Adobe Systems Software Ltd, Ireland), with living (leaf) and dead material (twigs < 5 cm thick) noted at each grid intercept around the outer 15 cm edge of the canopy (Fig. S1). From the total number of intercepts (dead + live), the percentage of canopy dieback was calculated for each of the two photographs and the result averaged per tree.

Statistics

The data were split into three sets according to river and species: A. haematoxylon and A. erioloba in the Auob; and A. erioloba in the Nossob.

The differences between upstream and downstream for six dependent variables were assessed: Wilcoxon Sum Rank tests for leaf δ13C values, linear mixed effects models for δ18O and δ2H values and midday xylem pressure potentials, linear models for specific leaf area and Kruskal–Wallis Rank Sum tests followed by a Pairwise Wilcoxon with Bonferroni correction for canopy dieback. The models were developed using ‘season’ (wet/dry), ‘position’ (upstream/downstream) and the interaction between these with individual trees as the random effect (the same trees were sampled in both seasons). All tests on the data were assessed in R©v3.1.2 (R Development Core Team, 2014) and a value of p < 0.05 required for significance.

Results

Water source

Borehole water level

Instrument malfunction for our piezometers affected recordings, so that only those readings of which we are certain were considered (Fig. S2). Water level depth varied between 38 m to 46 m in the Auob and 49 m to 59 m in the Nossob. In both the Auob and Nossob, the water table at the upstream borehole was lower than that of the downstream borehole. This was unexpected, but we speculate that this is as a result of a calcrete layer close to the surface underlying the aquifer for the downstream boreholes in both rivers. Both the Auob boreholes show a similar pattern in groundwater flux through time with a steady drop in water level of ±4 m soon after the peak of the dry season (July/August), and a subsequent rise of ±4 m matching the peak of the wet season rains (January/February). The Nossob downstream water level showed a 6 m drop two months after the peak of the dry season, but only a 2 m rise during the peak of the following wet season with an unusually low asymptote from the middle of the wet season that may be the result of instrument malfunction (Fig. S2).

Oxygen and hydrogen stable isotopes

Meteoric waters worldwide follow a Rayleigh distillation process that results in a linear relationship between δ18O and δ2H, termed the global meteoric water line (GMWL; y = 8x + 10) (Craig, 1961; Gat, 1996). For plant water source studies in arid environments these linear relationships (essentially representing evaporation) are extremely useful as deep (non-evaporatively enriched) and shallower moisture sources (evaporatively enriched) can be readily distinguished, with deep water plotting more negative values and shallow water more positive values (February, West & Newton, 2007; West et al., 2012). We constructed our own local meteoric water line (LMWL; y = 5.06x − 6.75) from rain water samples collected at Twee Rivieren in January and August and between North Kwang and Nossob in August 2013 (Fig. 2). Rainfall at our study site is extremely seasonal and even after no rainfall for several months and with mean midday temperatures between 30 °C and 40 °C our study trees were not deciduous with δ18O and δ2H values indistinguishable from that of groundwater (Fig. 2).

Figure 2 Mean δ18O and δ2H values (±1 SE) for xylem water of (A) Auob River, Acacia haematoxylon (B) Auob River, Acacia erioloba and (C) Nossob River, Acacia erioloba for two seasons (wet and dry).

Values are plotted relative to the local meteoric water line (y = 5.06x − 6.75) with rain (+) and mean values for borehole water (BH) included.

The results for all xylem water samples for both A. erioloba and A. haematoxylon in the Auob River plotted below the LMWL with wet season values similar to that of groundwater (Fig. 2). In the dry season, however, downstream of the abstraction point, xylem water δ18O values for both species are significantly different from groundwater δ18O values (Fig. 2). In the Nossob, δ18O values are similar to that of the Auob in the wet season in that xylem water δ18O values are indistinguishable from that of the groundwater (Fig. 2). Conversely, in the dry season, δ18O values for xylem water in trees both upstream and downstream of the abstraction borehole are significantly different from groundwater values. The downstream values are however more positive than the upstream values.

Plant moisture stress

Leaf stable carbon isotope ratios

In the Auob, there were no significant differences in leaf δ13C values for either upstream or downstream A. haematoxylon or A. erioloba (Fig. 3). In the Nossob however, A. erioloba values downstream (−24.7 ± 0.4‰) were significantly more enriched (W = 52, p = 0.04) relative to upstream δ13C values (−26.3 ± 0.37‰).

Figure 3 Mean leaf δ13C values (±1 SE) for (A) Auob River Acacia haematoxylon, (B) Auob River Acacia erioloba and (C) Nossob River Acacia erioloba.

Different letters indicate significant differences at p < 0.05 (Wilcoxon Rank Sum Test, unpaired) between upstream and downstream.

Xylem pressure potentials

In the Auob, there were no significant differences in midday xylem pressure potentials (XPP) for either upstream or downstream trees (Fig. 4). In the Nossob the downstream trees in the dry season (average −3.1 MPa) had significantly more positive XPP’s than upstream trees regardless of season (average −3.5 and −3.4 MPa respectively).

Figure 4 Mean midday xylem pressure potentials (±1 SE) for both wet (dark) and dry (light) seasons for (A) Auob River Acacia haematoxylon, (B) Auob River Acacia erioloba and (C) Nossob River Acacia erioloba.

Different letters denote significant differences at p < 0.05 (linear mixed effects model, Simultaneous tests for generalised linear hypotheses) between upstream and downstream trees within and between seasons.

Specific leaf area

In the Auob River, both tree species downstream of the active borehole had significantly lower SLA values in the dry season than in the wet season. For the Nossob River, SLA values were overall significantly lower in the dry season than in the wet season, but there was no significant difference between upstream and downstream (Fig. 5).

Figure 5 Mean specific leaf area (±1 SE) for both wet (dark) and dry (light) seasons for (A) Auob River Acacia haematoxylon, (B) Auob River Acacia erioloba and (C) Nossob River Acacia erioloba.

Different letters denote significant differences at p < 0.05 (ANOVA, Tukey HSD) between upstream and downstream within and between seasons.

Canopy dieback

There were no significant seasonal differences in average percentage canopy dieback for A. haematoxylon in the Auob and A. erioloba in the Nossob (Fig. 6). In the Auob, A. erioloba showed significantly more dieback downstream in both seasons. Average canopy dieback was higher in A. haematoxylon (between 26 and 30%) than in A. erioloba (between 12 and 20%).

Figure 6 Mean percentage canopy dieback (±1 SE) for both wet (dark) and dry (light) seasons for (A) Auob River Acacia haematoxylon, (B) Auob River Acacia erioloba and (C) Nossob River Acacia erioloba.

Different letters indicate significant differences at p < 0.05 (Kruskal–Wallis Rank Sum test followed by a Pairwise Wilcoxon with Bonferroni correction) between upstream and downstream.

Discussion

The depth to the aquifer in the two rivers at our study site varies between 38 m and 59 m with our water isotope analysis demonstrating that the trees in these rivers are using this deep water. While ours is the first study to demonstrate the use of deep groundwater for A. haematoxylon these results are in agreement with research 120 km south east of our study site in the Kuruman River showing this for A. erioloba (Schachtschneider & February, 2013). Our results also show that there are differences between the low water use site in the Auob and the high water use site in the Nossob. In the Auob both of our study species use deep ground water in the wet season. In the dry season, however, these trees use deep groundwater upstream of the borehole and in losing contact with the groundwater use more isotopically enriched soil water downstream. Significantly lower specific leaf area values for both species downstream indicate that in the dry season these downstream trees have physiologically adapted to the change in available water by producing smaller heavier leaves with marginally but not significantly more negative XPP’s.

As in the Auob, A.erioloba in the Nossob use deep ground water in the wet season. In the dry season, however, all trees, both upstream and downstream, use more isotopically enriched soil water and have smaller heavier leaves. Significantly more positive δ13C values for downstream trees suggest that these trees are maintaining leaf water potential through stomatal closure, a conclusion verified by more positive XPP’s (McDowell et al., 2008). These results demonstrate that trees in the Nossob adjust to lower levels of water availability by closing stomata to maintain relatively high water potentials thereby preventing loss of or damage to xylem conducting tissue as is indicated in the low percentages of canopy dieback (Jones & Sutherland, 1991; West et al., 2012). With only a change in SLA but no change in XPP’s or leaf δ13C values and little canopy dieback, A.erioloba in the Auob do not demonstrate the same levels of stress to changes in available water.

At our study site in the Nossob near a borehole with a higher extraction rate than at our study site in the Auob, the A. erioloba trees demonstrate physiological adaption to a flux in the water table of between 4.5 m and 5.2 m between the wet and the dry season. This flux in water table depth is similar to the 5 m flux attributed to water abstraction in the north-western sector of South Africa (Van Dyk et al., 2008). In the Sonoran desert of the American South West, groundwater declines of 18–30 m have resulted in low stem water potentials, reduced leaf size and high levels of canopy mortality in Prosopis velutina (Stromberg, 1993). Similarly, in the Murray Darling Basin of SE Victoria in Australia, declines in water level of between 12.1 and 22.6 m for Eucalyptus camaldulensis and 12.6 and 26.7 m for Eucalyptus populnea resulted in significantly poorer canopy condition (Kath et al., 2014). Our trees in the Nossob are exhibiting similar responses with a much smaller fluctuation in the water table.

Trees in the Kalahari are the deepest rooted trees in the world and our results show that the reason for these deep roots is to source water at depths of between 40 and 60 m (Canadell et al., 1996). While the Nossob trees may be physiologically tolerant of 4–5 m fluctuations in water table, this could change with drought such as the current 2015–2016 one.

Conclusion

The higher abstraction for the Nossob camp has resulted in the trees near that camp showing signs that groundwater depths in the dry season have exceeded thresholds identified by studies in Australia and America as being critical (Kath et al., 2014; Stromberg, 1993). The decline in water table in the Kalahari is not as much as that in Australia and America but the water column in the Kalahari trees is under greater negative pressure because of the distance to the water table of 40–60 m as opposed to 5 m in the American SW (Koch et al., 2004; Ryan & Yoder, 1997). The increase in dry season tension on the water column resulting from a decline in the water table has resulted in the Nossob trees controlling for water loss through stomatal regulation. However, in the event of a drought, such stomatal regulation can result in carbon starvation, canopy dieback and canopy death (McDowell et al., 2008; West et al., 2012). Our perception is that there are many more dead trees in the Nossob than there are in the Auob. We did not evaluate this but we speculate that these deaths were the result of some past drought event.

National Parks not only in South Africa but throughout the world are under pressure to increase revenue. This growing demand for ecotourism will increase pressure on groundwater resources in arid systems where there are no alternative water sources. Aquifer management may be described as the art of abstracting no more water than is replenished. Such abstraction should take into consideration the ecological reserve set out in the National Water Act (Republic of South Africa, 1998) that allows for an adequate and timeous supply of water to maintain the integrity of not only the rivers but also the terrestrial plants reliant on the aquifer (Baron et al., 2002). There has been no published research on abstraction or replenishment rates for any of the aquifers supplying the rest camps in the Park. Our results would suggest very strongly that this should be a research focus for the future and that the Park should develop a strategic adaptive management approach for groundwater use. Such an approach would develop thresholds for potential concern (Biggs & Rogers, 2003) that would allow for an adequate ecological reserve as our study suggests that trees in the Park, and in particular the Nossob, are nearing and have possibly exceeded the threshold of physiological tolerance.

Supplemental Information

Figure S1 Illustration of the grid overlay and intercept counting method implemented to determine canopy dieback from photographs

Click here for additional data file.

Figure S2 Average monthly groundwater depths for the Auob (circles) and Nossob (triangles) Rivers

Open circles are Kamqua picnic site; closed circles, Urikaruus Water hole; open triangles, Quibit’je Quap; and closed triangles, North Kwang. Open symbols are downstream and closed symbols are upstream.

Click here for additional data file.

We would like to thank South African National Parks for allowing us to do the research in the Kgalagadi Transfrontier Park. We are grateful to Graeme Ellis, Paola Vimercati and Amy Betzelberger for help with the fieldwork.

Additional Information and Declarations

Competing Interests

Author Contributions

Data Availability

The authors declare there are no competing interests.

Eleanor Shadwell conceived and designed the experiments, performed the experiments, analyzed the data, wrote the paper, prepared figures and/or tables, reviewed drafts of the paper.

Edmund February conceived and designed the experiments, performed the experiments, analyzed the data, contributed reagents/materials/analysis tools, wrote the paper, prepared figures and/or tables, reviewed drafts of the paper.

The following information was supplied regarding data availability:

South African National Park Data Repository: http://dataknp.sanparks.org/sanparks/metacat/judithk.111141.4/sanparks.

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
