# Peer review of "Effects of groundwater abstraction on two keystone tree species in an arid savanna national park"

_PeerJ, doi:10.7717/peerj.2923_

## Round 0.1 · original submission · Minor Revisions

Please, address each of the suggestions and remarks of the reviewers to revise your manuscript.

Reviewer 1 ·

Basic reporting

General comments:
In several places kms is used instead of km units.
Abstract:
Line 23: replace “above” and “below” with “upstream” and “downstream” (also in other parts of the text).
Line 26: rephrase “that this is the same water that is being abstracted” as it seems repetitive; you’re comparing with water that is being abstracted. You may want to introduce the concept of antagonism for the same resource both here and in the background section of the abstract.
Lines 35-36: Are there any pumping tests to demonstrate the amount of increase in abstractions that would lead to a draw down that would be hazardous? Otherwise “Any increase” may be too strong here. I think it’s sufficient to establish the existence of risk.
Introduction:
The authors reference locations that may be unknown to the reader. Furthermore, tree locations relevant to boreholes, etc. are mentioned. In order to have a more clear view of the case study and to be in a position to adequately assess the experimental design, a figure including as many relevant features as possible is required.
Furthermore, the two first paragraphs of the introduction have a mixed scope, the third provides a flavor of the methodology and the last gives some insight on the relevance and impact of the scientific problem, something also mentioned earlier. I suggest organizing this part better, providing the reader with background information on vegetation characteristics under such environments, documented behaviors that you will later try to capture with the described methods, etc. (e.g. in lines 77 isotopes are mentioned but no references to other studies are provided), in well-defined paragraphs. Many case specific elements can be transferred to the study site section.
Some more specific comments on the introduction include:
Lines 46-50: Consider rephrasing as “and alternatively” and “This despite” confuse the meaning of the sentence. I understand that Nel et al. (2007) estimate that Auob and Nossob are not threatened. Nevertheless the cited paper does not specifically mention these rivers. Can you explain? A map, as suggested earlier could help.
Lines 49-63: This part sounds like an excommunication of infrastructure development. Mention specifically “non-sustainable infrastructure development”. Also, there are surely studies addressing this issue, at both theoretical (e.g. Shah et al., 2007; note the term “extinction depth” which may also be part of the discussion here) and practical level (e.g. Sommer and Froend, 2011).
Lines 67-69: Parts referring to a comparison of the case study with Schachtschneider and February (2013) can be left out here. If the comparison is valid it will be demonstrated in the results and it can be discussed later. Maybe a more general statement can be given here, e.g. that there are cases such as those described in Schachtschneider and February (2013) where action needs to be taken, without referring to the current case study specifically.
Methods
Data: As far as I could see, groundwater levels are not available online, nevertheless the figure provided in the supplement is sufficient.
References:
Shah, N., Nachabe, M. and Ross, M. (2007), Extinction Depth and Evapotranspiration from Ground Water under Selected Land Covers. Ground Water, 45: 329–338. doi:10.1111/j.1745-6584.2007.00302.x
Sommer, B. and Froend, R. (2011), Resilience of phreatophytic vegetation to groundwater drawdown: is recovery possible under a drying climate?. Ecohydrol., 4: 67–82. doi:10.1002/eco.124
Schachtschneider, K. & February, E.C. Plant Ecol (2013) 214: 597. doi:10.1007/s11258-013-0192-z

Experimental design

As mentioned above, a figure showing the case study, boreholes, rivers, and sampled trees would greatly assist in evaluating the experimental design.
In the “water source” section very little information is provided regarding the nature of the aquifer, its hydrogeological characteristics, the amounts of water abstracted, etc. It is possible that if these characteristics were known they could justify later findings (e.g. lines 205-207: “this was unexpected”).
Regarding the methods followed for experimental design, the background section should provide more information about relevant literature where they have been used and standard values in comparable studies. In most of the relevant method sections (e.g. leaf stable isotope ratios, xylem pressure potentials, specific leaf area, canopy dieback) there is a single reference which in some cases is not enough to back the hypothesis behind using this method. For example, in the “specific leaf area” section, Ackerly et al. (2002) is cited for specific soil moisture vs specific leaf area but no direct reference to such findings could be found there. Something more relevant to the behavior of SLA in drought conditions may be found in Liu and Stützel (2004). I suggest revisiting these sections.
Regarding the “Oxygen and hydrogen stable isotope” section and Figure 1, it would help discussion of the global meteoric water line was included in the diagrams. I understand that here you’re trying to form an argument for deep rooted plants using groundwater during the wet season and other sources of moisture during the dry season. Have any shallow rooted species been investigated for the same property? Assuming they don’t have access to groundwater they would have different xylem water characteristics during the wet season and similar during the dry season.
References
F. Liu, H. Stützel, Biomass partitioning, specific leaf area, and water use efficiency of vegetable amaranth (Amaranthus spp.) in response to drought stress, Scientia Horticulturae, Volume 102, Issue 1, 15 October 2004, Pages 15-27, ISSN 0304-4238, http://dx.doi.org/10.1016/j.scienta.2003.11.014.

Validity of the findings

Lines 269-270: The adaptation of the downstream trees may be only speculation unless similar findings can be cited.
Conclusions can be enriched.

Reviewer 2 ·

Basic reporting

The article reports a very interesting study on consequences of groundwater abstraction on two tree species in an arid savanna national park. It is well written. The introduction could be improved by referencing studies that have looked at groundwater abstraction impacts on vegetation, perhaps also in other regions. Consistency in tekst/figures would help. Now for example groundwater/borehole water is used.

Experimental design

For the reader it would help to include a map of the area with sampling locations, borehole and the location of the calcrete layer. Include how many twig samples were collected per tree. The piezometers that failed: in time, or in certain locations(?), it is not clear right now.

Validity of the findings

What was the p-level of the significant findings?

Additional comments

The conclusion does not include the findings of the paper so much, but instead lists the broader consequences. It would be nice if the findings are included in the conclusion.

---

## Round 0.2 · Minor Revisions

The paper has been improved in line with the suggestions and remarks of the reviewers. Only a couple of small issues still need some additional attention, see below for more details. I trust that the follow-up by the authors can be quick in order to speed up the publication process thereafter.

Reviewer 1 ·

Basic reporting

Overall the authors have improved the manuscript. They still need to review newly added parts for syntax. Regarding their rebuttal, point 7 (Lines 46-50: Consider rephrasing as “and alternatively” and “This despite” confuse the meaning of the sentence.) was not faced. I appreciate that the authors disagree with the comment, on the other hand, I am confused by these two sentences and believe they could be worded more clearly.

Experimental design

no comment

Validity of the findings

In their rebuttal the authors acknowledge several limitations (e.g. no pumping tests to demonstrate the trend of pumping abstractions, lack of a geological map or other research on the aquifer). I believe that these limitations need to be highlighted in the text, possibly to underline the importance of their methods in the absence of other evidence, or the need to expand applied research to the direction collecting the required evidence. Conclusions reached by the authors need to be formulated in the view of these limitations, especially in the newly written first paragraph of the conclusions part.

---

## Round 0.3 · accepted · Accept

The last minor issues have now been resolved and addressed accordingly. Well done!